# A Crossed Pure Agraphia by Graphemic Buffer Impairment following Right Orbito-Frontal Glioma Resection

**DOI:** 10.3390/ijerph20021346

**Published:** 2023-01-11

**Authors:** Eva M. Arroyo-Anlló, Claudette Pluchon, Coline Bouyer, Vanessa Baudiffier, Veronique Stal, Foucaud Du Boisgueheneuc, Michel Wager, Roger Gil

**Affiliations:** 1Department of Psychobiology, Neuroscience Institute of Castilla-León, University of Salamanca, 37005 Salamanca, Spain; 2Neurology Department—Neuropsychology Unit, Poitiers University Hospital, 86021 Poitiers, France; 3Clinical Electrophysiology Department, Poitiers University Hospital, 86021 Poitiers, France; 4Neurosurgery Department, Poitiers University Hospital, 86021 Poitiers, France; 5Neurology Department, Poitiers University Hospital, Poitiers University, 86021 Poitiers, France

**Keywords:** tumour, glioma, frontal, right hemisphere, language, graphemic buffer

## Abstract

Pure agraphias are caused by graphemic buffer damage. The graphemic buffer stores graphemic representations that handle the transition from spelling lexicon to writing or oral spellings. The authors report a case of a crossed pure agraphia, following the post-surgical removal of a right frontal low-grade glioma in a right-handed French patient. He presented a pure agraphia displaying the features of a graphemic buffer impairment. Our patient only made spelling errors, whereas repetition and other oral language abilities remained perfect. We found a greater number of errors for longer stimuli, increased errors for the medially located graphemes, and agraphia for both words and non-words and error types, essentially consisting of omissions, substitutions, and letter transpositions. We also observed no significant effect of word frequency on spelling errors, but word length affected the rate of errors. The particularity of this case was linked to right frontal subcortical injuries in a right-handed subject. To our knowledge, it is the first report of a crossed pure agraphia caused by graphemic buffer impairment. Further studies are needed in order to analyse the role of subcortical structures, particularly the caudate nucleus in the graphemic buffer during writing tasks, as well as the participation of the non-dominant hemisphere in writing language.

## 1. Introduction

Pure linguistic agraphia is a rare disorder, characterized by an isolated impairment of writing that is not accompanied by any other language impairment or impairment in praxis [1]; thus, it is different from pure apraxic agraphia and spatial agraphia. Pure linguistic agraphia is the result of a disruption to central linguistic processes involved in writing, but verbal language and reading abilities are intact. Nevertheless, there may be semantic and/or orthographic errors in writing with a normal calligraphy quality. Case-based studies suggest that pure linguistic agraphia may be further subdivided into phonological and lexical subtypes [2,3,4]. In addition, Caramazza and his team [5] described a new type of agraphia characterized by a deficit of letters assembly/association and a preserved orthographic system, due to a graphemic buffer syndrome. A graphemic buffer is a component of the working memory system whose role is maintaining the order and identity of abstract letter identities [6,7]. It is assumed that errors at this level should reflect the difficulties of moving from orthographic lexicon to writing or to oral spelling, which requires the temporary storage of graphemic representations in a graphemic buffer of a specialized working memory [8]. There are few reported cases of pure agraphia in the literature due to the impairment of graphemic buffer [9,10,11,12,13].

Studied cases of brain injuries involved in pure agraphia focused on various brain regions, but most of them described injuries in the left hemisphere of right-handed patients, in particular the left middle frontal gyrus [11,13,14,15,16,17,18,19]. To our knowledge, only a case of crossed kana agraphia was reported following an infarct in the Wernicke’s area of the non-dominant cerebral hemisphere [20].

Caramazza et al. [5,6] described both deep and superficial pre- and post-left rolandic injuries in a case of an agraphia-compromised graphemic buffer. Several responsible injuries of pure agraphia due to graphemic pure impairment have been found, but they were all in the dominant cerebral hemisphere. Although there are few cases of pure agraphia due to graphemic buffer deficit, the injuries reported include a left fronto-parietal region and basal nuclei [10], a right putaminal haemorrhage in a left-handed patient [11] and a left frontal abscess [13]. However, to our knowledge, no cases of crossed pure agraphia suggesting graphemic buffer impairment after non-dominant cerebral hemispheric injuries have ever been described in the literature.

We herein report a case of a crossed pure agraphia, following a post-surgical removal of a right frontal low-grade glioma in a right-handed French patient. He presented a pure agraphia displaying the features of a graphemic buffer syndrome. To our knowledge, this is the first report of a crossed pure agraphia caused by graphemic buffer impairment.

## 2. Case Report

### 2.1. History, Neuro-Imaging and Surgical Treatment

The patient JR was 52-year-old, right-handed French carpenter, assessed using the Humphrey laterality questionnaire [21]. He had no past family history of left-handedness. He had received 12 years of formal education. There was no history of neurological illness or developmental learning disorder in childhood.

In March 2010, he suffered a first generalized tonic–clonic seizure from the outset, revealing an adult diffuse low-grade glioma diagnosed by a brain MRI in July 2010. The patient was operated on under general anaesthesia in August 2010, and the whole glioma was removed, but in March 2012, he had another generalized seizure. The June 2012 MRI disclosed a recurrence of the tumour. At that time, awake brain surgery had become the standard of care in our university hospital at Poitiers (France), particularly for low-grade gliomas [22], in order to optimize oncological benefits and minimize the functional risk of brain tumour removal, as well as achieving supra-total removal. In June 2012, the patient underwent a second surgical removal under local anaesthesia, and supra-total removal was achieved. The awake brain surgery enabled us to check the lack of participation of the right hemisphere in oral language capacities, which were never affected at any stage of the procedure. Unfortunately, we did not check the written language capacities considering the localization of glioma (right cerebral hemisphere) in this right-handed patient due to no previous family history of left-handedness.

In the post-operative period, there were no complications. Anti-seizure medication and close monitoring were carried out, as well as cerebral MRIs at regular intervals until January 2015, and the patient resumed their normal professional and personal life. The last MRIs, made after 31 months (January 2015) from the awake brain surgery, disclosed no tumour regrowth (Figure 1). In addition, we used a postoperative diffusion tensor imaging (DTI), which is an MRI technique that depicts the integrity of white matter tracts, detecting arcuate fasciculus (Figure 2) and inferior occipito-frontal fasciculus bilaterally.

### 2.2. Neuropsychological Evaluation: Pre-Operative and Post-Operative Neuropsychological Assessments

A comprehensive set of neuropsychological tests was administered to the patient before and after the awake brain surgery to evaluate the cognitive capacities. Global cognitive functions were evaluated using Rapid Evaluation of Cognitive Function (RECF) [23]. We also used a DO80 verbal naming test [24], which assessed picture naming to evaluate language abilities. Different aspects of executive function were assessed with the Stroop test [25], the frontal assessment short test (FAST) [26] and the Wisconsin Card Sorting Test (WCST) [27]. In addition, we also used the verbal fluency tasks as the category fluency task [28], (animals as items) and letter fluency [29], (items beginning with letter /m/). Tests measuring visual–spatial and constructive abilities included the WAIS–III block design [30] and the Rey–Osterrieth complex figure (ROCF) copying test [31]. Verbal working memory was assessed using RECF subtest of digit span forward and backward [23] and WAIS-III digit span subtest [30].

For the pre-operative neuropsychological examination before the awake brain surgery (June 2012), the results of every cognitive test were within the normal range (Table 1).

For the post-operative neuropsychological examination (Table 1), the outcomes of neuropsychological test set remained within the normal range three months after awake brain surgery (October 2012). Later, the same neuropsychological assessment was made in March 2014 (eighteen months post-operative), observing similar results to that of October 2012, though the scores were lower for letter fluency and WAIS–III Block design subtest. Furthermore, we also observed some difficulties with spelling certain words during this second neuropsychological assessment follow-up; therefore, we decided to carry out a thorough investigation of oral and written language, using the French adaptation of the Boston Diagnostic Aphasia Examination (BDAE) [32].

In March 2014, a language examination found that oral language abilities are within normal scores (Table 2). The conversational and expository components of speech were preserved, and the patient’s spontaneous speech was fluent. Oral expression was conserved, and his auditory comprehension was normal. Patient JR was able to pronounce words correctly or repeat single words and sentences, and he showed no dysarthria. The score for picture naming was a normal value for his age. Patient JR also showed preserved reading (words/sentences) and reading comprehension.

Concerning written language abilities, automatic writing (name and address), dictated written transcription of the alphabet in upper or lower cases, as well as copying out a sentence were normal (Table 2). By contrast, numerous errors were observed during written transcription of words under dictation (e.g., Sportif: “*sorptif*”). Patient J.R. was aware of his mistakes, and he always tried to self-correct them (ex. Spectacle: “*septacle, secpta, sceptacle*”). Exact oral repetition of every dictated stimulus was required before the tests and was always accurate, followed by written transcription or spelling. These errors mostly consisted of graphemic substitutions (58.88%), omissions (13.33%), and transpositions (22.44%), associated with a lower number of graphemic insertions (3.35%). This assessment of written language was then repeated twice, in May and, later, October 2014. A similar distribution pattern of error types was also found in May and October 2014 (substitutions: 59.33% and 31.08%; omissions: 25% and 25.84%; transpositions: 16.66% and 31.08%; insertions: 9.11% and 11%, respectively). The assessments of the transcription of words written under dictation were caried out in March, May and October 2014 and are shown in detail in [App app-ijerph-20-01346]. The word list was relatively balanced based on the word frequency (F) (half the word list with F < 500 and the other half with F > 500) from the Brulex database [33]. We observed that errors were not affected by words frequency (*p* = 0.35, 0.44 and 0.58 in March, May, and October 2014). However, we found that word length, particularly words of more than six letters, had an effect (p < 0.01 in March, May, and October 2014). The patient made more than 24% of their errors from words with more than six letters (Table 3). Consequently, we examined positions of literal mistakes for each erroneous transcription of dictated words, using the procedure of Caramazza et al. [6] to normalize the distribution of literal errors for stimuli of various lengths. Thus, we analysed three letter “positions” for the wrong transcription of each stimulus: at the beginning (position 1), at the middle (position 2), and at the end of the word (position 3). The distribution of errors for words as a function of letter position in a stimulus is depicted in Figure 3. We observed that most mistakes concerned mid-word graphemes, and very few were located at the beginning or the end of a word (Figure 3). Furthermore, similar errors using several logatomes were observed during writing under dictation (e.g., “*pralclame, fralclame or fraclame*” instead of fraglame, or “strusclame” instead of strublag), as well as during the oral spelling of stimuli with more than six letters.

Considering these spelling errors, we extensively completed the cognitive assessment in October 2014, using the Wechsler Adult Intelligence Scale (WAIS-IV) [30] and the Wechsler Clinical Memory Scale (WMS-III) [34]. Patient JR obtained a normal intellectual quotient of 91, considering full scale, as well as normal scores for Verbal Comprehension Index scale (score: 88) and Perceptual Reasoning Index (score: 98). Nevertheless, we found that Processing Speed Index (score: 81) and the reverse order digit span (score: 3) were a little lower than expected. However, the rest of the Working Memory Index subtests were within normal performances. Moreover, memory capacities of patient JR remained normal using WMS-III, including auditory memory (score: 99), visual memory (score: 85), visual working memory (score: 85), immediate memory (score: 90), and delayed memory (score: 86), although delayed visual memory index (score: 80) was a little lower than other memory abilities.

## 3. Discussion

In this paper, we presented a case of a right-handed patient who developed a crossed pure linguistic agraphia, whose features suggested an impairment of graphemic buffer, following right frontal low-grade glioma surgical removal. To our knowledge, our patient is the first case of crossed pure linguistic agraphia caused by a graphemic buffer deficit to be described in the scientific literature.

Agraphia is an impairment or loss of a previous ability to write. It often occurs concurrently with other neurologic deficits, but pure agraphia cases are rare [1]. Functional cognitive models of writing production distinguish between two levels of writing mechanisms: central and peripheral levels. The central level belongs to the language functional architecture per se and involves a two-route feature [3], with a sublexical processing (operating the conversion of phonemes into graphemes) and lexical processing (translating auditory word forms to their written counterparts). Linguistic agraphia is caused by the impairment of these central mechanisms, and is characterized by writing of well-formed letters, while spelling errors dominate. By contrast, peripheral or apraxic agraphia is observed when agraphia relates to impairment of the peripheral level of writing processes, characterized by agraphic deficits, mainly involving allography, grapheme or letter shaping.

The features of this patient’s pure agraphia cannot be attributed to the peripheral phases of writing, but a deficit of central cognitive processes, particularly a graphemic buffer impairment. The symptomatology of this patient was determined postoperatively, during a task consisting of the written transcription of words under dictation. Educational level can have an effect on spelling errors, but we did not find a significant difference between errors as a function of different word frequencies. Our patient’s work was more manual, essentially requiring visuospatial and calculation skills. As far as we know, he never noticed the impairment of his writing, nor did his family/friends notice any writing deficits, even when writing text messages or social media content. However, social media makes it possible to use more flexible spelling and grammar rules and uses applications that automatically correct spelling or lexical errors. Unfortunately, a limitation of our study was that it did not assess writing capacities before surgery, which would lead to better understanding of this case. Nevertheless, a recent systematic database search found that dysgraphia appeared post-operatively in 26.9% of the cases and persisted at follow-up in approximately half [35].

Our patient only made spelling errors, whereas repetition and other oral language abilities remained perfect. Most errors were graphemic substitutions, omissions, and transpositions. Similar kinds of errors were also made in the oral and written spelling of words. We observed no significant effect of word frequency on spelling errors. Graphemic buffer functioning is traditionally considered not to be influenced by word frequency, although the review by Sage and Ellis [36] showed that the influence of frequency on the graphemic buffer are not exceptional on spelling. In contrast, we found an impact of word length and graphemes’ serial position on writing words and no-words. Errors were more common for longer words and were more numerous in the middle of the writing stimuli. In 1987, Caramazza [6] et al. reported a case of agraphia, whose features suggested a selective deficit of the graphemic buffer following a vascular injury in pre- and post-rolandic areas (both superficial and deep) in the left hemisphere. Our patient showed the same profile of spelling errors as those in the case from Caramazza et al. [5,6]: greater number of errors for longer stimuli; increased errors for the medially located graphemes; agraphia for both words and non-words and error types consisting of substitutions, deletions, transpositions, or insertions of letters. They suggested that the graphemic buffer plays a crucial role in transiently storing graphemic representations before their conversion into specific letter shapes. Our patient showed normal scores in verbal and visual working memory; therefore, there was no global working memory disorder. In addition, oral spelling was of course affected, as were all other modes of written language production. However, the absence of any repetition deficit did not allow us to conclude that the phonological output buffer was affected. Additionally, writing errors made by our patient remained relatively stable, both in qualitative and quantitative terms, throughout post-operative follow-ups. Few studies have described pure agraphia/dysgraphia with similar difficulties in letters assembly and no other language impairment, suggesting a selective impairment of the graphemic buffer involved in spelling processes troubles [9,10,11,12,13]. Nevertheless, the two cases of Schiller and his team [12] found that spelling errors most often occurred at the end of words, proposing potentially different types of impairment to the graphemic buffer [37]. In addition, Wing and Baddeley [38] found that a predominance of spelling errors toward the end of a word may reflect a rapid deterioration of graphemic representations. Further studies of pure agraphia with graphemic buffer impairment could effectively explore impairments to components of the graphemic buffer, in order to increase our understanding of different processes in the working memory system and how these impact on writing language.

Most neuroimaging studies related to agraphia in right-handed patients found that brain lesions were located in the left hemisphere, but there is still no consensus on the neural basis of agraphia [15,18,39,40]. Neural correlates of pure agraphia due to graphemic buffer deficit reported in the literature also remain unclear, although were mostly caused by left hemisphere injuries in right-handed patients. Only the case of Kokubo et al. [11] reported a left-handed patient who presented a pure agraphia caused by an impairment of the graphemic buffer, following a right putaminal haemorrhage. Several cerebral injuries have been found, including the supramarginal gyrus and the dominant parietal lobe [41], the angular gyrus and intraparietal sulcus [42] and fronto-parietal injuries [6,10]. In addition, graphemic buffer impairments were associated with subcortical damage to prefrontal areas and in pre- and post-central gyri [43], as well as basal nuclei [10,11]. Therefore, it appears that cortical as well as subcortical injuries could be held responsible for deficits of the graphemic buffer, playing an important role in pure agraphias.

Our right-handed patient suffered a right orbitofrontal injury, which was extended to the head of the right caudate. These injuries of non-dominant cerebral hemispheric caused a crossed pure agraphia due to graphemic buffer impairment. Moreover, post-operative DTI tractography bilaterally detected arcuate fasciculus and inferior fronto-occipital fasciculus. Little is known about the subcortical pathways involved in writing language. Motomura et al. [44] described the first case of transient agraphia and alexia elicited through intra-operative direct subcortical stimulation, and they found that the implicated tract was the deep portions of the dorsal inferior fronto-occipital fasciculus. In addition, subcortical damage to prefrontal areas and in pre- and post-central gyri were associated with graphemic buffer impairments [43]. Additionally, two large meta-analyses [39,45] observed that the angular gyrus was not independently identified in central spelling processes.

Awake brain surgery in our patient verified the absence of right hemisphere involvement in language. Indeed, neither pre-operative cortical nor subcortical stimulations elicited language disorder at any step of the procedure. Post-operatively, we observed a pure agraphia compatible with an impairment of graphemic buffer. The graphemic buffer is a component of working memory that holds orthographic representations active, while the spelling process takes place [6]. Recent review studies found that the prefrontal cortex is critical for the brain network of working memory system, connected to the cortico-subcortical frontal loops [46]. As we indicated above, the cases published up until now were either caused by deep injuries (basal ganglia), or from cortical but also subcortical damages, but further studies are needed to analyse the role of basal ganglia in the graphemic buffer during writing tasks, as well as the participation of the non-dominant hemisphere in writing language. Moreover, it would be useful to pay more attention to spelling assessments in pre-, intra- and post-operative awake surgery practices, in order to better understand current neurofunctional theories of spelling, as well as to facilitate the return to work and maintain quality of life.

## 4. Conclusions

We report a case of a crossed pure linguistic agraphia in a right-handed patient, following right frontal low-grade glioma surgical removal. Our patient only made spelling errors, which were attributed to a writing deficit of the central cognitive process, in particular a selective impairment of the graphemic buffer. To our knowledge, this patient is the first described case of crossed pure linguistic agraphia due to graphemic buffer deficit.

Further studies are needed to explore the impact of working memory processes on writing language, as well as the role of non-dominant hemisphere and subcortical areas in the graphemic buffer during writing language, particularly the basal ganglia.

## Figures and Tables

**Figure 1 ijerph-20-01346-f001:**
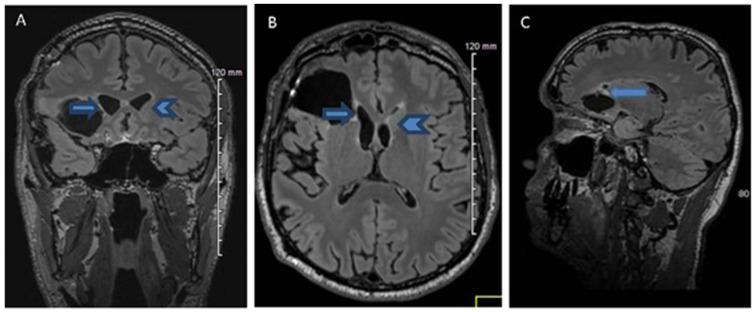
MRIs from MRI 31 months (January 2015) following the awake brain surgery: FLAIR Frontal (**A**), FLAIR Axial (**B**) and FLAIR Sagittal (**C**). Views showing supra-total resection with no visible recurrence of the tumour: head of left caudate (**A**,**B**: arrow head); head of right caudate was partially resected (**A**–**C**: arrow).

**Figure 2 ijerph-20-01346-f002:**
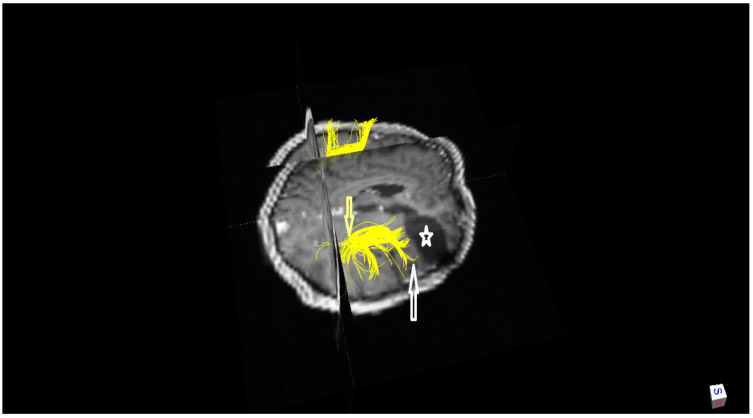
Magnetic Resonance Imaging (MRI) with diffusion tensor imaging (DTI) technique and fibre tracking (January 2015) revealed right arcuate fasciculi (yellow arrow). It also showed the anterior end of right arcuate fasciculus (white arrow), which is distant from the rear wall of the operative cavity (white star).

**Figure 3 ijerph-20-01346-f003:**
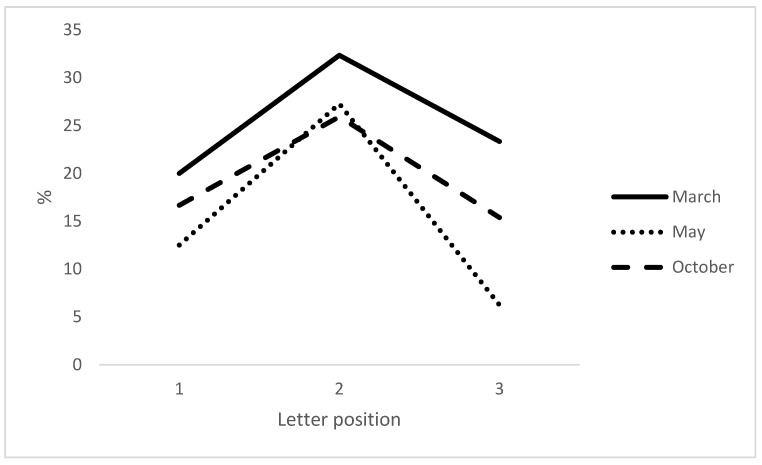
Distribution of errors in function of the letter positions (1, 2 and 3) during the assessments in March, May, and October 2014.

**Table 1 ijerph-20-01346-t001:** Results of pre-operative (in June 2012) and post-operative neuropsychological assessments (in October 2012, March and May 2014).

Test	June 2012 Pre-Operative Evaluation	October 2012 Post-Operative Evaluation	March 2014 Post-Operative Evaluation	May 2014 Post-Operative Evaluation
RECF	47/50 (NP)	46.5/50 (NP)	46.5/50 (NP)	46.5/50 (NP)
RECF digit span	1.5/2 (NP)	1.5/2 (NP)	2/2 (NP)	1.5/2 (NP)
WAIS-III digit span	7/19 (NP)	9/19 (NP)	9/19 (NP)	
DO80	79/80 (NP)	80/80 (NP)	79/80 (NP)	79/80 (NP)
FAST	18/18 (NP)	17/18 (NP)	15/18 (NP)	17/18 (NP)
WCST categories	6 (NP)	6 (NP)	6 (NP)	
WCST errors (%)	40% (NP)	0% (NP)	14.28% (NP)	
Stroop test (RS–PS)	0.08 (NP)	9.29 (NP)	4.85 (NP)	
Category fluency	15 (NP)	18 (NP)	17 (NP)	
Letter fluency	11 (NP)	14 (NP)	7 (NP)	
WAIS-III block design	11/19 (NP)	12/19 (NP)	9/19 (NP)	8/19 (NP)
ROCF copy type	IV (NP)	IV (NP)	IV (NP)	
ROCF copy time	169 (NP)	146 (NP)	163 (NP)	
ROCF copy total score	36 (NP)	33 (NP)	29	
ROCF recall type	II (NP)	IV (NP)		
ROCF recall total score	18 (NP)	17 (NP)		

RECF: Rapid Assessment of Cognitive Functions; DO80: DO80 verbal naming test; FAST: frontal assessment short test; WCST: Wisconsin Card Sorting Test; RS-PS: real score minus predictive score; ROCF: Rey–Osterrieth complex figure; NP: non-pathological.

**Table 2 ijerph-20-01346-t002:** Results of the French adaptation of the Boston Diagnostic Aphasia Examination (BDAE).

BDAE Subtests	Score	BDAE Subtests	Score
**Spontaneous oral expression**		**Aphasic transformations of oral language**	
- Conversation, free narration, image description	7/7	- Phonemic paraphasias, morphologic verbal, neologisms	0
		- Semantic verbal paraphasias	0
**Repetition**		- Syntagmic paraphasias	0
- Words	10/10		
- Concrete sentences	8/8	**Written language comprehension**	
- Abstract sentences	8/8	- Letters and words visual recognition	10/10
		- Words recognition	8/8
**Denomination**		- Spelled words recognition	8/8
- Contextual denomination	30/30	- Match between image and word	10/10
- Images denomination	105/105	- Sentences and texts understanding	10/10
- Body parts denomination	30/30		
		**Writing**	
**Reading aloud**		- Graphism	3/3
- Words	30/30	- Automatic writing (letters of the alphabet, numbers from one to ten)	36/36
- Sentences	10/10	- Sentence copying	8/8

**Table 3 ijerph-20-01346-t003:** Number of stimuli transcription errors and percentages under dictation in the function of stimulus length during the assessments in March, May, and October 2014.

Stimulus Length	March 2014		May 2014		October 2014	
	Word Errors	Non-Word Errors	Word Errors	Non-word Errors	Word Errors	Non-Word Errors
3–6 letters	0/9 (0%)	0/1 (0%)	0/14 (0%)	0/1 (0%)	0/17 (0%)	0/4 (0%)
7–9 letters	4/11 (36.36%)	2/2 (100%)	7/29 (24.13%)	3/3 (100%)	13/54 (24.07%)	2/4 (50%)
10–13 letters	5/12 (41.66%)		8/17 (47.05%)		11/33 (33.33%)	

## Data Availability

All material is available at the departments of Neurology and Neurosurgery of University Hospital, CHU La Milétrie, Poitiers, France.

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
