# Peer review of "A Crossed Pure Agraphia by Graphemic Buffer Impairment following Right Orbito-Frontal Glioma Resection"

_ijerph, 2023, doi:10.3390/ijerph20021346_

Round 1

Reviewer 1 Report

A Crossed Pure Agraphia by Graphemic Buffer Impairment following Right Orbito Frontal Glioma Resection

The authors report a very interesting single-case description of a person with pure agraphia after right orbito frontal glioma resection.

The case description first gives a comprehensive theoretical overview of pure agraphia and reports in detail the linguistic and cognitive examinations of the patient. Some critical information is not (yet) reported and should complete the case description. I will address these topics below.

Line 92: It is reported that the patient suffered pure agraphia as a result of his second surgery. It is described that he was able to resume his social and professional life after the surgery successfully. It is remarkable that the patient did not notice the impairment in writing himself. Even if a carpenter doesn't have to write that much when doing his job, the problems should be noticeable when writing, for example, text messages on social media. At the very least, this point should be raised and discussed as a limitation in the discussion.

Line 107 ff: When describing the neuropsychological test battery, it is noticeable that no measures of verbal working memory (e.g. digit span forward and backward) are reported. Especially because the graphemic output buffer is considered to be part of / or involved in the working memory, such measures would be crucial. If such tests were performed as part of the assessments performed (RACF?), it would be essential to report these subscores. At least the neuropsychological assessments in 2014 should have been supplemented by tests of working memory.

Line 129: It is reported that abnormalities in (oral?) spelling in the March 2014 examination led to the extended examination of oral and written language skills, and comparable errors occurred in oral spelling as in writing (line 209). Although the performance in repetition (and thus the phonological output buffer?) showed normal performance according to the authors, at least a possible involvement of the phonological output buffer in the oral spelling tasks should be discussed.

Line 138: It is said that "automated writing" (name and address) were not affected, but that many errors occurred when writing from dictation. The detail These are described in qualitative detail, but it would also be important to describe how many words were written in each of the word lengths listed. Please add these figures.

The informative appendix gives information about the qualitative and quantitative error types and the length effect is evident. Especially, however, the orthography of the long words appears complex. Can it be excluded that the occurring errors were additionally caused by an education-specific influence? Considering that the spelling errors were not noticed in the patient's everyday life, this possibility should at least be discussed.

Although I am a non-native speaker of English, intensive language editing of the manuscript seems necessary to me. 

Author Response

Firstly, we thank to the reviewers for your constructive suggestions. We’ve added some critical information to complete the case, such as:

  1. Line 92.

You’re completely right. Unfortunately, we didn´t check his writing capacities before surgery, but we’ve included your important remark in the discussion.

  1. Lines 107 and 129.

Yes, we agree with you. The RECF also assessed the verbal working memory using the subtest of digit span forward and backward and WAIS-III digit span. We’ve included the scores in the table 1, which were normal in all evaluations.

Grapheme Buffer Syndrome involves a specialized working memory. The neuropsychological examination of our patient showed no impairment of visual or verbal working memory.

The impairment of the phonological output buffer impairs repetition. The latter was normal in our patient, which does not allow us to retain such an impairment. In these conditions, it is legitimate to think that the impairment of oral spelling is due solely to the impairment of the graphemic buffer, which effectively disrupts all modes of written language production (handwriting, typing, oral spelling, mobile letters, etc.).

We’ve also included in the discussion.

  1. Line 138.

We’ve included a Table showing how many words were written in each of the word lengths listed (Table 3).

  1. You’re right. Thus, we tried to avoid educational effect considering the word frequency in the word list. The word list was relatively balanced based on the word frequency –F- (half the word list with F < 500 and the other with F > 500) from Brulex database [33].

We’ve added that in the text, too.

  1. Lucy Mainah Wanja made the English checking of manuscript. She is a Professional translator (wanja2193@gmail.com).

Reviewer 2 Report

Re-formulate the sentence in the lines 86-88.

Line 94. The Figures are generally cited in the format: Fig.1.

Figure 2. Please provide more details in figure's caption, including how you obteined this reconstruction ( the figure should be understandable stand-alone).

In the table 1 I suggest to clarify (maybe in the first line) which assessment has been performed before the surgery and which assessment has been performed after the surgery. Moreover I suggest to include more details in the Table's caption (the table should be understandable stand-alone) .

Line 189 "in isolation1". Please, re-formulate this sentence.

Line 205 "persisted at 205 follow-up in approximately half [34]." I suggest to re-formulate this sentence.

Line 237 "Most neuroimaging studies on writing brain regions observed that left hemisphere 237 was involved in writing processes of right-handed subjects, but no consensus about the 238 neural circuits involved [15,18,38,39]." I suggest to re-formulate this sentence.

Line 258 "impairments47". Maybe you should add the brackets.

Author Response

Firstly, we thank to the reviewers for your constructive suggestions.

  1. Diffusion-tensor imaging (DTI) is a magnetic resonance imaging (MRI) technique that depicts the integrity of white matter tracts. Magnetic resonance image –MRI- with Diffusion Tensor Imaging technique (DTI) and fiber tracking (January 2015) can detect subtle white matter changes not normally seen on conventional MRI can be detected.

We’ve explained in the text the DTI technique in the text and we‘ve modified the figure’s caption to describe the image.

Reviewer 3 Report

This paper was very well written, clear, and intriguing. I found only a few typo or minor semantic problems. The fist line of the abstract is missing a word. The abstract includes the sentence,  "We also observed no significant effect of words frequency on spelling errors,  but word length." This is incomplete. Did you mean "We also observed no significant effect of words frequency on spelling errors, but word length affected the rate of errors." 

Line 189 says " Our patient is reported is the first case..." Please clarify.

Line209 says "aloud words produced similar kind of errors." Did you mean aloud words produced a similar kind of errors"?

Author Response

Firstly, we thank to the reviewer for your constructive suggestions.

  1. We’ve put the missing word in the first line of the abstract.
  2. Yes, you’re right. We’ve added your sentence in abstract.
  3. We’ve modified the sentences in line 189 and 209

Round 2

Reviewer 1 Report

Dear authors,

thank you for addressing all comments. The text is much improved and I have no further comments on the content. Although I do not presume to judge the correctness of the language, it seems to me that in some places the language is not yet scientific and technical enough. Some unusual technical terms are used. For example, instead of "aloud spelling", "oral spelling" is usually used. In Table 2: "Aloud reading" should become "reading aloud". Also, in the discussion, which has become much more sophisticated in content, some sentences seem relatively informal (e.g., on page 8 "He was a carpenter...").

In addition, the formatting of the tables still appears unfinished (Table 1: word separation, columns, line breaks; Table 2: figures partly centered, partly not).

Thanks for revisiting the remaining comments.

Author Response

Dear colleague

Thank you again for your useful suggestions.

  • We’ve changed the terms (aloud, aloud reading…) you detected in the text and the table 2.
  • We’ve also modified the sentence on page 8.
  • We’ve tried to improve the tables 1, 2 and 3.
  • And finally, the professional translate has checked English in all text, again.
